# Redox-Dependent Modulation of Human Liver Progenitor Cell Line Fate

**DOI:** 10.3390/ijms24031934

**Published:** 2023-01-18

**Authors:** Francesco Bellanti, Domenica Mangieri, Giorgia di Bello, Aurelio Lo Buglio, Giuseppe Pannone, Maria Carmela Pedicillo, Alberto Fersini, Michał Dobrakowski, Aleksandra Kasperczyk, Sławomir Kasperczyk, Gianluigi Vendemiale

**Affiliations:** 1Department of Medical and Surgical Sciences, University of Foggia, 71122 Foggia, Italy; 2Department of Clinical and Experimental Medicine, University of Foggia, 71122 Foggia, Italy; 3Department of Biochemistry, Faculty of Medical Sciences in Zabrze, Medical University of Silesia, 41-800 Katowice, Poland

**Keywords:** redox balance, HepaRG, liver regeneration

## Abstract

Redox homeostasis is determinant in the modulation of quiescence/self-renewal/differentiation of stem cell lines. The aim of this study consisted of defining the impact of redox modifications on cell fate in a human hepatic progenitor line. To achieve this, the HepaRG cell line, which shows oval ductular bipotent characteristics, was used. The impact of redox status on the balance between self-renewal and differentiation of HepaRG cells was investigated using different methodological approaches. A bioinformatic analysis initially proved that the trans-differentiation of HepaRG toward bipotent progenitors is associated with changes in redox metabolism. We then exposed confluent HepaRG (intermediate differentiation phase) to oxidized (H_2_O_2_) or reduced (N-acetylcysteine) extracellular environments, observing that oxidation promotes the acquisition of a mature HepaRG phenotype, while a reduced culture medium stimulates de-differentiation. These results were finally confirmed through pharmacological modulation of the nuclear factor (erythroid-derived 2)-like 2 (NRF2), a principal modulator of the antioxidant response, in confluent HepaRG. NRF2 inhibition led to intracellular pro-oxidative status and HepaRG differentiation, while its activation was associated with low levels of reactive species and de-differentiation. In conclusion, this study shows that both intra- and extracellular redox balance are crucial in the determination of HepaRG fate. The impact of redox status in the differentiation potential of HepaRG cells is significant on the utilization of this cell line in pre-clinical studies.

## 1. Introduction

Stem cells rely on their capacity to maintain pluripotency, self-renew, or activate/differentiate to promote regeneration on intricate signals in their niches. Characterized by special regenerative properties, the liver is able to respond to different stimuli, activating several pathways for mass restoration: indeed, acute organ loss induces hepatocellular proliferation characterized by phenotypic fidelity (canonical pathway), while chronic hepatic injury may activate the trans-differentiation of parenchymal cells or activation of facultative progenitors (alternative pathways) [1,2]. In particular, this latter regenerative mechanism involves a bipotent cell line located in the smallest branches of the ductular tree (regenerative niche), capable of differentiating toward hepatocyte or biliary cell lineages [3,4,5,6].

To study regeneration and differentiation, the HepaRG cell line was used as an alternative to progenitors showing human oval ductular bipotent characteristics [7,8]. Indeed, confluent HepaRG cells supplemented with dimethyl sulfoxide (DMSO) are stimulated to differentiate toward hepatocytes [7,9]. Notably, the great plasticity of HepaRG is demonstrated by the ability of differentiated cells to retro-differentiate toward progenitors when seeded at low density [8].

Several liver diseases are characterized by persistent damage that promotes overproduction of reactive species and disruption of redox balance [10]. Loss of redox homeostasis leads to hepatocellular senescence and impairs liver regeneration [11]. Alteration of redox balance may impact the activity of several transcription factors involved in the modulation of quiescence, self-renewal, and differentiation of stem cell lines [12]. We previously demonstrated that models of liver progenitor activation show hepatic oxidative stress, which is able to activate nuclear factor (erythroid-derived 2)-like 2 (NRF2), the master regulator of the antioxidant response, in liver parenchyma but not in the regenerative niche [13]. Indeed, we observed that NRF2 inhibition triggers activation and differentiation of liver progenitors, and we were able to determine the commitment of HepaRG into functionally differentiated hepatocytes [13]. In a subsequent investigation, we described that senescence-induced mitochondrial dysfunction may limit the differentiation capacity of HepaRG, suggesting the involvement of redox perturbations in the biology of this cell line [14].

In this study, we aimed to characterize the effect of redox status on the balance between self-renewal and differentiation of HepaRG cells and to define the impact of NRF2 activation.

## 2. Results

### 2.1. Trans-Differentiation of HepaRG toward Bipotent Progenitors Is Associated with Changes in Redox Metabolism

With the aim of defining the main redox-dependent pathways that are associated with HepaRG trans-differentiation, we analyzed publicly accessible gene expression data sets from untreated and DMSO-treated cells using Gene Expression Omnibus 2R (GEO2R) and gene set enrichment analysis (GSEA; GEO accession numbers GSE112123 [15] and GSE181963 [16]). Enrichment scores of data sets of untreated versus DMSO-treated cells revealed that the upregulation of metabolism signals and downregulation of pathways associated with cell proliferation occurred during the differentiation process (Figure 1a). In both data sets, assessment of gene ontology categories that were significantly upregulated (GSEA: family-wise error rate *p* < 0.05) in trans-differentiated cells were related to oxidoreductive activity (Figure 1b).

Taken together, bioinformatic analysis strongly suggests that oxidoreductive activity increased in DMSO-treated HepaRG cells. This observation is further supported by a higher NAD^+^/NADH ratio and production of fluorescent reactive species in DMSO-treated as compared with undifferentiated HepaRG cells (Appendix A). 

### 2.2. Impact of Redox Status Modification on HepaRG Differentiation

To analyze the effects of redox manipulation on cell differentiation, confluent HepaRG were exposed to redox-active chemicals. To boost the oxidative level, HepaRG cells were cultured with H_2_O_2_ (simplest peroxide, strong oxidizer, second messenger). On the contrary, to expose cells to a reduced environment, the medium was complemented with N-acetylcysteine (NAC, a cysteine prodrug that triggers glutathione synthesis). The acute toxicity of each compound was tested, and the maximum non-toxic concentration was utilized in successive experiments (Appendix A).

Of note, 5 days of treatment of HepaRG with H_2_O_2_ enhanced the bipotent phenotype (hepatocyte-like cells characterized by larger size and polygonal shape, including markedly round nuclei and biliary-like cells), while NAC treatment caused a reduction in cell size and morphological changes consistent with de-differentiation toward basal HepaRG (Figure 2).

No significant effect of redox status perturbation was observed on HepaRG apoptosis (Appendix A); of interest, H_2_O_2_ exposure inhibited the cell cycle by increasing the number of cells in the G0/G1 state and reducing those in the G2/M state, while NAC treatment resulted in activation of the cell cycle as suggested by the increased quantity of cells in S and G2/M state (Figure 3).

Finally, with respect to vehicle-treated cells, H_2_O_2_ exposure resulted in higher expression of albumin, cytochrome P450 3A4 (CYP3A4), and gamma glutamyl transpeptidase 1 (GGT1) genes (characteristic of differentiated cells), and lower expression of carcinoembryonic antigen (CEA) and cytokeratin 19 (CK19) genes (representative of immature cells); on the contrary, NAC treatment reduced the expression of albumin, CYP3A4, and GGT1 genes, and increased CEA expression (Figure 4).

Taken together, these results suggest that a pro-oxidative environment promotes acquisition of a mature HepaRG phenotype, while a reduced culture medium stimulates de-differentiation of confluent cells. 

### 2.3. Different NRF2 Pharmacological Modulations Induce Opposite Effects on HepaRG Differentiation

In a previous study, we described that undifferentiated HepaRG treated with ARE expression modulator 1 (AEM1), a chemical NRF2 inhibitor, acquired the phenotype of trans-differentiated cells [13]. Thus, to study the impact of NRF2 signaling modulation on bipotent cell differentiation, we exposed confluent HepaRG to AEM1 or CPUY192018 (a powerful inhibitor of NRF2–Keap1 interaction, with consequent activation of the NRF2 pathway). The acute toxicity of both chemicals was assessed to determine the maximum non-toxic dosage (Appendix A).

With respect to control cells, HepaRG treated with AEM1 enhanced their biopotency by acquiring a mixed population phenotype, characterized by biliary-like cells surrounding hepatocyte-like colonies; this morphology was not observed in cells treated with CPUY192018 (Appendix A). Of note, the production of fluorescent reactive species in confluent HepaRG cells was boosted by AEM1 but reduced by CPUY192018 (Appendix A). Compared with vehicle-treated cells, AEM1 treatment reduced the S phase of the cell cycle, while CPUY192018 exposure enhanced the G2/M phase and reduced the G0/G1 phase (Appendix A). Furthermore, AEM1 treatment increased the expression of albumin, CYP3A4, and GGT1 genes, and reduced CEA gene expression; the opposite effect was observed in HepaRG cells treated with CPUY192018 (Appendix A). The expression of adhesion molecules such as CD49a (associated with hepatocyte-like cells) and CD49f (limited to biliary-like cells) was higher in AEM1-treated and lower in CPUY192018-treated HepaRG with respect to vehicle-treated cells (Figure 5). Similarly, the expression of CD184/C-X-C motif chemokine receptor 4 (a classical endoderm marker) and of EpCAM (marker of hepatic progenitor cells) was enhanced by AEM1 and reduced by CPUY192018 (Figure 5).

Taken together, these results indicate that NRF2 inhibition may incentivize the differentiation of bipotent HepaRG, while NRF2 activation may de-differentiate confluent cells.

## 3. Discussion

The present study demonstrates that redox status is determinant in the regulation of human liver progenitor cell-like fate. HepaRG cell self-renewal is maintained by low levels of reactive species, while increased production/exposure to oxidants leads to the loss of proliferation activity and to the acquisition of mature cell phenotypes. To confirm this hypothesis, our study design considered different approaches, which indicated that both intra- and extracellular redox homeostasis may influence stemness/differentiation of progenitors.

HepaRG is an exclusive cell line that shows great expression of transport proteins, metabolizing and detoxifying enzymes, and nuclear receptors, but also has a unique ability to trans-differentiate toward biliary-like and hepatocyte-like cells [7,8,17]. More than being a feasible instrument for drug metabolism, virology, and cell biology studies, differentiated HepaRG cells have also been used in rodents to create humanized liver, with the aim of studying hepatic physiology and development in vivo [18,19]. The potency of HepaRG to behave as progenitors has been morphologically and phenotypically characterized, classifying three phases (proliferative, intermediate, and differentiated) during a 4-week culture period [7,9]. The proliferative phase is shown during the first days of culture, but cell confluence after 2 weeks triggers commitment into both hepatocyte-like and biliary-like cells (intermediate state), which can further mature after DMSO treatment (differentiated state). Our bioinformatic analysis, based on previous transcriptomic data sets from undifferentiated and differentiated HepaRG cells, shows modifications in the gene level of several pathways related to oxidoreductive metabolism (increased expression) and cell cycle (reduced expression) during differentiation induced by cell confluence followed by DMSO exposure. These changes are associated with increased NAD^+^/NADH redox cycle and oxidant production, and this result is consistent with a metabolic shift toward aerobic glycolysis/tricarboxylic acid cycle/oxidative phosphorylation [20]. DMSO is an amphiphilic compound regularly used as a solvent for small hydrophobic drug molecules and for cell cryopreservation [21]. DMSO, at low concentrations, acts as a free radical scavenger, but can be pro-oxidant at higher concentrations [22]. In the present study, we exposed HepaRG cells to 2% DMSO according to previously published differentiation protocols [7,8]. Even though the definition of a direct effect of DMSO on HepaRG redox status was beyond the scope of this work, we are confident that the redox perturbations described are mostly associated with changes in oxidative metabolism, rather than the applied concentration of DMSO. Moreover, this hypothesis is consistent with previous data showing higher mitochondrial activity in HepaRG cells undergoing the differentiation protocol [14]. It is strongly conceivable that this metabolic shift might lead to increased production/exposure to intracellular reactive species, as required to exit stemness and prime differentiation [23].

Stem cells reside in unique anatomic sites characterized by extremely specialized microenvironments (named as niches), in which peculiar intercellular and cell–matrix communications and signaling are maintained to modulate self-renewal and differentiation processes [24]. Of note, stem cell niches warrant a stable mild oxygen tension, exposing progenitors to low extracellular concentrations of reactive species [25]. A pro-oxidative microenvironment may lead to the loss of stemness and trigger differentiation in several stem cell lines, and this effect can be neutralized by reducing agents [26,27,28,29]. Our results confirm such observations in the hepatic progenitor-like cell line HepaRG at intermediate phase of differentiation, because we observed that exposure to non-toxic amounts of hydrogen peroxide as an extracellular oxidizing agent leads to loss of stemness features and stimulates differentiation (inhibition of cell cycle, and increased expression of both biliary and hepatocellular differentiation markers), while the addition of a reducing compound in the culture medium accounts for proliferation and de-differentiation.

Hydrogen peroxide is the most important messenger in the redox-dependent modulation of biological processes [30,31]. H_2_O_2_ production is triggered by metabolic signals, but also by chemokines, growth factors, or physical stressors [32]. H_2_O_2_ may regulate several signal transduction factors involved in cellular and developmental mechanisms. Of note, H_2_O_2_ activates the master regulator of antioxidant response, NRF2, via multiple mechanisms [33]. In turn, NRF2 activates cytoprotective signaling, counteracting oxidative damage through the expression of genes encoding antioxidants and phase II detoxifying enzymes [34]. NRF2 is activated in alcoholic, non-alcoholic, viral, and toxic hepatitis, supporting liver protection from oxidative injury [35]. Moreover, growing evidence supports a novel function of NRF2 signaling in stem cell fate [36]. We previously described the involvement of NRF2 in hepatic progenitor homeostasis because its inhibition promotes activation and differentiation [13]. In this study, after proving that the pharmacological activation and inhibition of NRF2 determine opposite changes in intracellular levels of oxidants, we confirmed that blocking NRF2 promotes differentiation, but we also found that its stimulation leads to de-differentiation. Investigation of the molecular pathways associated with NRF2 modulation was beyond the aim of our study, but it will represent an interesting topic for future research. 

In conclusion, the present report demonstrates that both intra- and extracellular redox balance are crucial in the determination of hepatic progenitor-like HepaRG cell line fate, because exposure to a pro-oxidative environment leads to loss of proliferation and triggers differentiation, while a reduced status promotes stemness and quiescence. Additional studies are needed to address the molecular pathways underlying our conclusions. The impact of redox status in the differentiation potential of HepaRG cells, with consequent changes on their regenerative and metabolic features, may have significant consequences on the utilization of this cell line in pre-clinical studies.

## 4. Materials and Methods

### 4.1. Cell Line, Cultures, and Treatments

The human cell line HepaRG was obtained by Merck Millipore (MMHPR116, Merck KGaA, Darmstadt, Germany). Undifferentiated HepaRG cells show a fibroblast-like morphology, and the differentiation protocol at confluence stimulates both biliary-like epithelial and hepatocyte-like phenotypes, indicating bipotent progenitor characteristics [8]. HepaRG cells were seeded at 27,000 cell/cm^2^ confluence in a base medium composed of William’s E Medium + GlutaMAX (3255-020, Thermo Fisher Scientific, Waltham, MA, USA), supplemented with 10% FBS (F7524, Merck KGaA, Darmstadt, Germany), 100 U/mL penicillin (13752, Merck KGaA, Darmstadt, Germany), and 100 μg/mL streptomycin (P4333, Merck KGaA, Darmstadt, Germany). Medium was replaced once every 3 days and cells were passaged after each week. To obtain HepaRG differentiation, a previously described 2-step process was applied: cells were first cultured in the medium for 2 weeks and then treated with 2% dimethyl sulfoxide (DMSO, 276855, Merck KGaA, Darmstadt, Germany) for 2 more weeks [8].

Confluent HepaRG cells (intermediate differentiation state) were treated with pro-oxidants and antioxidants put in the culture medium every 24 h for 5 consecutive days. To render cells more oxidized, they were exposed to increasing amounts (from 1 nM to 100 µM) of H_2_O_2_; to make cells more reduced, N-acetylcysteine (NAC, A7250, Merck KGaA, Darmstadt, Germany) was added to the medium. ARE Expression Modulator 1 (AEM1, SML-1556, Merck KGaA, Darmstadt, Germany) was utilized to impede NRF2 binding to antioxidant response elements (AREs). CPUY192018 (GLXC-08803, GLIXX Laboratories, Southborough, MA, USA) was used to inhibit NRF2–Keap1 interaction. Both AEM1 and CPUY192018 were dissolved in growth medium, which was added as the vehicle in control cells. Each experiment was performed in triplicate.

The microculture tetrazolium assay was used to test toxicity of each substance after 24 h [37], and the maximum non-toxic dosage was considered in successive experiments.

### 4.2. Intracellular NAD^+^/NADH Content

A NAD/NADH-Glo Bioluminescent Assay kit (G9072, Promega Corporation, Madison, WI, USA) was used to measure total intracellular NAD^+^/NADH, following the manufacturer’s guidelines. After lysis with dodecyltrimethyl ammonium bromide (DTAB), HepaRG cells were treated to neutralize their counterparts. For NAD^+^ quantification, the extract was treated with 25 μL of 0.4 N HCl and heated at 60 °C for 15 min; then, 25 μL of Trizma base was added before incubation at room temperature for 10 min. To measure NADH, the extract was first incubated at 60 °C for 15 min and then at room temperature for 10 min; 50 μL of HCl/Trizma solution was added to the extract. A reductase reduced a pro-luciferin reductase to luciferin in the presence of each species. The light intensity (relative to the quantity of each metabolite) was measured with a luminometer (DTX 880 Microplate Reader, Beckman Coulter, Brea, CA, USA).

### 4.3. Flow Cytometry Analysis

Reactive species production, apoptosis, cell cycle, and phenotype changes were investigated by flow cytometry analysis, using a FlowSight Cytometer (Amnis, Merck KGaA, Darmstadt, Germany) and IDEAS Software [13].

Intracellular levels of reactive species were monitored using 2′,7′-dichlorofluorescein diacetate (DCF, 35845, Merck KGaA, Darmstadt, Germany). HepaRG cells at a density of 2 × 10^5^ cells/well were seeded in a six-well plate and incubated overnight. The cells were then treated with DCF (5 μM) at 37 °C for 30 min.

Apoptosis was analyzed in HepaRG cells seeded in six-well plates and stained with Annexin V-FITC (5 µL for one sample)/7-amino-actinomycin D (7-AAD, 5 µL for one sample) (FITC Annexin V Apoptosis Detection Kit with 7-AAD, 640922, BioLegend, San Diego, CA, USA) for 30 min at room temperature, and then 500 µL of binding buffer was added before flow cytometry analysis.

To analyze the cell cycle, cells were washed with PBS and centrifuged at 300× *g* for 3 min; supernatant was discarded, pellet was resuspended in the medium, and cells counted. Then, cold EtOH was added, and samples were vortexed and stored overnight at −20 °C. The following day, cells were centrifuged at 300× *g* for 3 min and stained with 5 μM DRAQ5 (424101, BioLegend, San Diego, CA, USA) for 15 min at room temperature.

To study cellular phenotype, PE-labeled antibodies against CD49a (1:50, 130-101-397), CD49f (1:50, 130-097-246), CD184 (1:10, 130-098-354), EpCAM (1:50, 130-091-253), and rat IgG1ĸ isotype control (1:10, 130-102-645) were purchased from Miltenyi Biotec, Bergisch Gladbach, Deutschland. In brief, 1.0 × 10^6^ cells/sample were resuspended in PBS and incubated in the dark for 10 min at 4 °C; cells were then centrifuged at 300× *g* for 10 min and washed twice with PBS. Samples were finally resuspended in PBS before flow cytometry analysis.

### 4.4. Gene Expression Analysis

To quantify gene expression, RNA was extracted from 1.0 × 10^6^ cells/sample; after reverse transcription, cDNA was used as a template for real-time polymerase chain reaction (PCR). RNA was extracted using a Pure Link RNA Mini kit (12183025, ThermoFisher Scientific, Waltham, MA, USA), following the manufacturer’s instructions. RNA concentration was spectrophotometrically quantified using the Nanodrop 2000/2000c (ThermoFisher Scientific, Waltham, MA, USA), detecting absorbance at λ = 260 nm. A260/A280 > 2 was considered to ensure protein-free samples. A High-capacity cDNA Reverse Transcription Kit (4368814, ThermoFisher Scientific, Waltham, MA, USA) was used for reverse transcription, and a fluorescent probe SYBR Green (172-5271, Bio-Rad Laboratories, Hercules, CA, USA) was used for PCR. Appendix A reports the sequences of forward and reverse primers of all the genes studied.

### 4.5. Bioinformatic Analysis

Microarray data from untreated and DMSO-treated HepaRG cells were analyzed for transcript expression using Gene Ontology (GO), Kyoto Encyclopedia of Genes and Genomes (KEGG), or gene set enrichment analysis (GSEA). Raw microarray data are also freely accessible on GEO using accession numbers GSE112123 [15] and GSE181963 [16]. are software was used to draw the Venn diagram [38].

### 4.6. Statistical Analysis

Data were represented as mean ± SEM of three independent experiments, with significance determined via Student’s unpaired t-test if comparing two groups, or one-way analysis of variance (ANOVA) when more than two groups were compared. Tukey’s multiple comparisons test was used as post hoc test for multiple comparisons. All statistics were performed using GraphPad Prism 6.0 software (San Diego, CA, USA). A *p* value of ≤ 0.05 was considered significant.

## Figures and Tables

**Figure 1 ijms-24-01934-f001:**
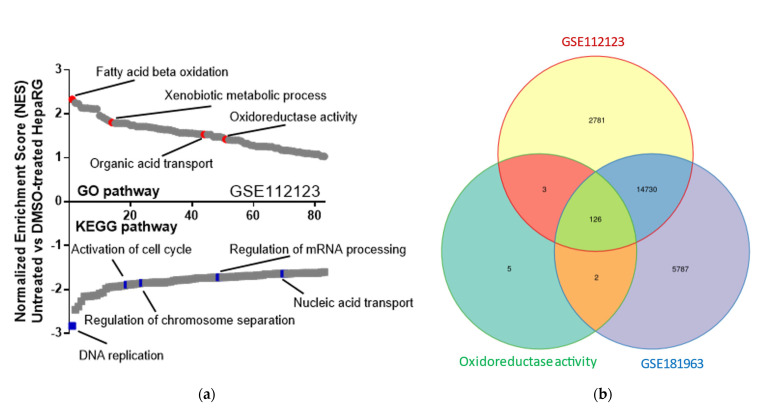
Oxidoreductive activity is induced by DMSO treatment and is associated with HepaRG trans-differentiation. (**a**) GSEA shows upregulated and downregulated signaling pathways in untreated HepaRG as compared with DMSO−treated cells. This panel represents evaluation results of microarray data from the freely accessible GEO data set (accession number GSE112123), using Kyoto encyclopedia of genes and genomes (KEGG) or Gene Ontology (GO) enrichment. Signaling pathways were classified on the basis of normalized enrichment scores (NESs); negative and positive NESs indicate upregulation or downregulation, respectively, in untreated HepaRG cells. Individual pathways related to metabolic or cell proliferation pathways are emphasized in red and blue, respectively. (**b**) Area−proportional Venn diagram showing 126 shared genes between the upregulated genes in transcriptomes deriving from DMSO−treated HepaRG cells (GSE112123 and GSE181963) and genes related to oxidoreductive activity (GO:0016491).

**Figure 2 ijms-24-01934-f002:**
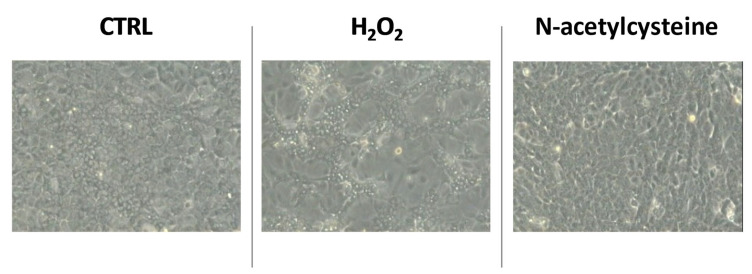
Representative microscopic images of confluent HepaRG cells exposed to vehicle (CTRL), 100 μM H_2_O_2_, or 1 mM N-acetylcysteine every 24 h for 5 days.

**Figure 3 ijms-24-01934-f003:**
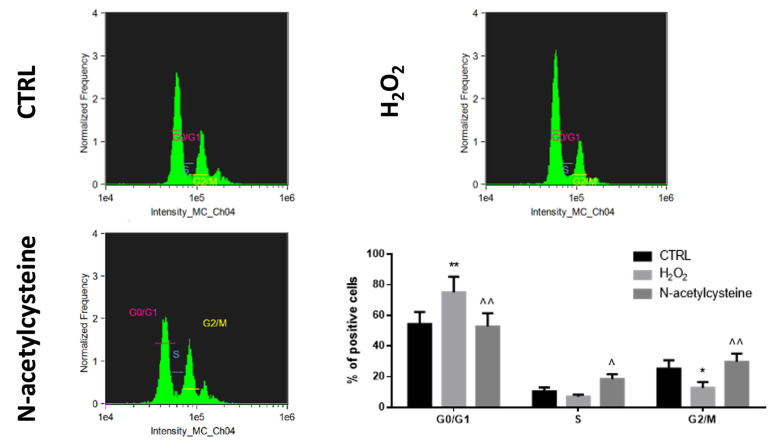
Cell cycle analysis in confluent HepaRG cells exposed to vehicle (CTRL), 100 μM H_2_O_2_, or 1 mM N-acetylcysteine every 24 h for 5 days. In the bar graph, data are illustrated as mean ± SEM of three replicates. Statistical differences were analyzed by one-way ANOVA; Tukey–Kramer was used as post hoc test. * = *p* < 0.05 vs. CTRL; ** = *p* < 0.01 vs. CTRL; ^ = *p* < 0.05 vs. H_2_O_2_; ^^ = *p* < 0.01 vs. H_2_O_2_.

**Figure 4 ijms-24-01934-f004:**
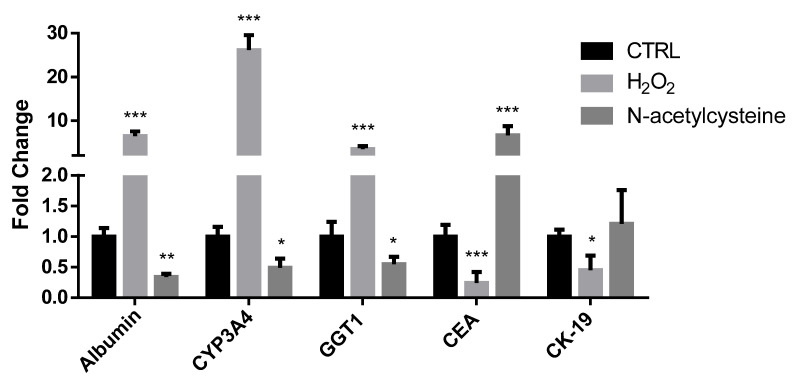
mRNA expression of genes associated with differentiation status in confluent HepaRG cells exposed to vehicle (CTRL), 100 μM H_2_O_2_, or 1 mM N-acetylcysteine every 24 h for 5 days. In the bar graph, data are illustrated as mean ± SEM of three replicates. Statistical differences were analyzed by one-way ANOVA; Tukey–Kramer was used as post hoc test. * = *p* < 0.05 vs. CTRL; ** = *p* < 0.01 vs. CTRL; *** = *p* < 0.001 vs. CTRL.

**Figure 5 ijms-24-01934-f005:**
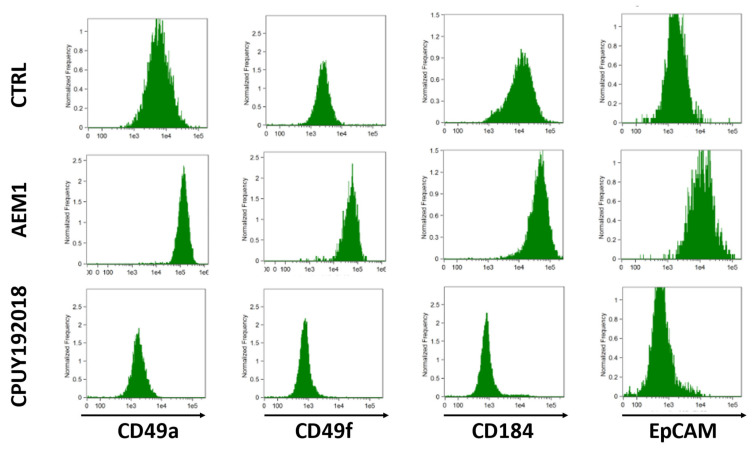
Flow cytometry histograms of confluent HepaRG cells exposed to vehicle (CTRL), 1 μM AEM1, or 10 μM CPUY192018 every 24 h for 5 days.

## Data Availability

The data presented in this study are available in the article or the Appendix A.

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
