# Peer review of "Redox-Dependent Modulation of Human Liver Progenitor Cell Line Fate"

_ijms, 2023, doi:10.3390/ijms24031934_

Round 1
Reviewer 1 Report
It is known that redox homeostasis is a determining factor in the modulation of rest/self-healing/differentiation of stem cell lines. In a peer-reviewed manuscript, the authors characterized the effect of redox status on the balance between self-renewal and differentiation of HepaRG cells using various methodological approaches. It has been shown that both intra- and extracellular redox balance is crucial in determining the fate of HepaRG. The effect of redox status on the differentiation potential of HepaRG cells is of great importance for the use of this cell line in preclinical studies.
The article is well written, the design of the study does not raise questions, the material is presented clearly and logically.
1. I did not find information about how many parallels the study was conducted in? Only in figure captions. Add to Materials and Methods
2. Small inscriptions on the drawings are hard to read. Please correct.
Author Response
It is known that redox homeostasis is a determining factor in the modulation of rest/self-healing/differentiation of stem cell lines. In a peer-reviewed manuscript, the authors characterized the effect of redox status on the balance between self-renewal and differentiation of HepaRG cells using various methodological approaches. It has been shown that both intra- and extracellular redox balance is crucial in determining the fate of HepaRG. The effect of redox status on the differentiation potential of HepaRG cells is of great importance for the use of this cell line in preclinical studies.
The article is well written, the design of the study does not raise questions, the material is presented clearly and logically.
Reply: we thank the reviewer for his positive comments on our study.
- I did not find information about how many parallels the study was conducted in? Only in figure captions. Add to Materials and Methods
Reply: the number of replicates was indicated in the Materials and Methods, as suggested (line 270, line 330).
- Small inscriptions on the drawings are hard to read. Please correct.
Reply: inscriptions on the drawings were modified (Figure 1 and Figure 3), as indicated.
Reviewer 2 Report
Thank you very much allow me to review the article entitle “Redox-dependent modulation of human liver progenitor cell line fate.” (ijms-2180236) that is presented for the Section Molecular biology in the Special Issue “Stem Cells in Health Diseases”.
The aim of this study was to characterize the effect of redox status on the balance between self-renewal and differentiation of HepaRG cells, and to define the impact of NRF2 activation. And they found that both intra- and extracellular redox balance are crucial in the determination of HepaRG fate. For this they conclude that the impact of redox status in the differentiation potential of HepaRG cells is significant on the utilization of this cell line in pre-clinical studies.
This is a work that is part of the research line of the group.
Comments:
In the abstract, the authors should indicate the objective of the study, as well as define "HepaRG" the first time it is used.
In the methodology, line 260 indicates that the cell was exposed, for how long were they exposed?
On line 286 “FlowSight Cytometer (Amnis, Merck KGaA, Darmstadt, Germany) and the IDEAS Software” should be referenced.
Author Response
Thank you very much allow me to review the article entitle “Redox-dependent modulation of human liver progenitor cell line fate.” (ijms-2180236) that is presented for the Section Molecular biology in the Special Issue “Stem Cells in Health Diseases”.
The aim of this study was to characterize the effect of redox status on the balance between self-renewal and differentiation of HepaRG cells, and to define the impact of NRF2 activation. And they found that both intra- and extracellular redox balance are crucial in the determination of HepaRG fate. For this they conclude that the impact of redox status in the differentiation potential of HepaRG cells is significant on the utilization of this cell line in pre-clinical studies.
This is a work that is part of the research line of the group.
Comments:
In the abstract, the authors should indicate the objective of the study, as well as define "HepaRG" the first time it is used.
Reply: the abstract was modified accordingly (lines 16-20).
In the methodology, line 260 indicates that the cell was exposed, for how long were they exposed?
Reply: cells were exposed every 24 hours for 5 consecutive days. This is now added in the methodology (lines 262-263).
On line 286 “FlowSight Cytometer (Amnis, Merck KGaA, Darmstadt, Germany) and the IDEAS Software” should be referenced.
Reply: we added a reference on line 289, accordingly.